# The Readiness of Lasem Batik Small and Medium Enterprises to Join the Metaverse

**Theresia Dwi Hastuti** [1,*], **Ridwan Sanjaya** [2] **and Freddy Koeswoyo** [1]

1 Accounting Department, Faculty of Economics and Business, Soegijapranata Catholic University, Semarang 50234, Indonesia
2 Information Systems Department, Faculty of Computer Science, Soegijapranata Catholic University, Semarang 50234, Indonesia
* Correspondence: theresia@unika.ac.id

**Abstract:** Today's business competitiveness necessitates the capacity of all company players, particularly small and medium enterprises (SMEs), to enter a broader market through information technology. However, the Lasem Batik SMEs have endured a great deal of turmoil during the COVID-19 pandemic. Marketing has been conducted through physical and internet channels, but the results have not been maximized. The purpose of this research was to consider the possibilities of Lasem Batik SMEs adopting metaverse technology as a marketing medium to enhance sales. The investigation was conducted on 40 Lasem Batik SMEs who met the requirements of using online media to sell their products, having a medium-sized firm, and displaying marketing that has reached the provincial level. The findings of this study are as follows: (1) The majority of participants stated that the metaverse is a virtual 3D space. This understanding is deepened by discussions about virtual 3D spaces that combine VR and AR, which today is often referred to as the metaverse. (2) Batik business owners hope that by using the metaverse, they will be able to obtain many benefits, especially related to market expansion. (3) Lasem Batik SMEs show great interest in expanding their marketing channels to a wider area; Lasem Batik entrepreneurs also accept the challenge of studying the metaverse with new knowledge and techniques they have never considered. (4) Overall, 75% of participants were ready to use the metaverse, and 25% still required guidance. (5) Local communities, universities, and large corporations provide great support for the use of the metaverse. (6) The commercial success of Lasem Batik SMEs is defined by product quality; ongoing online and offline advertising; originality and innovation; and the capacity to capitalize on possibilities, retain local wisdom, and preserve strong customer connections. The main conclusion is that the readiness of batik entrepreneurs to use the metaverse is highly dependent on the support of various parties. A strong desire to progress and develop one's business is the main factor determining one's intention to use the metaverse. As a result of the research, a prototype of a metaverse platform for Lasem Batik exhibitions has been developed. SMEs can use the room template provided by the platform and join other SMEs to hold a metaverse exhibition to attract global customers. These results can be connected to create a metaverse exhibition to attract global customers.

**Keywords:** augmented reality; marketing; metaverse; readiness; SMEs; virtual reality

## 1. Research Background

Over the last thirty years, immersive virtual reality (VR) and augmented reality (AR) technologies have continued to advance. Over the same period, network speeds have increased, culminating in the deployment of 5G mobile networks. Combined, these advances have greatly increased the prospects for the worldwide adoption of VR and AR. Facebook launched the metaverse and was followed by other platform providers with different names [1]. Facebook spent more than USD 10 billion in 2021 in developing metaverse technologies, and it is predicted that its investment will increase in the coming



years [2]. The metaverse is generally considered a network of 3D virtual worlds wherein people can interact, conduct business, and establish social relations through their virtual "avatars". It can be regarded as the virtual reality version of today's internet [3].

Over the past 18 months, public interest in the metaverse has soared, along with significant increases within companies [1]. The effects of the pandemic, especially in terms of restrictions on physical gatherings and travel, have spurred companies to seek new methods of working that can accommodate more authentic, cohesive, and interactive processes of remote and hybrid interaction and communication. The metaverse can address this need in four main ways: (1) new forms of immersive team collaboration; (2) the emergence of new digital partners that support artificial intelligence; (3) accelerated learning and skill acquisition through virtualization and gamified technology; (4) improving the economy through economic activities with metaverse media.

Widayani et al. [4] stated that today's business competition requires the capability of all businesses, especially SMEs, to penetrate a wider market and develop themselves by directing their business processes through technology and information advancements. The addition of online sales channels is an inevitable choice. The use of e-marketplaces is one of the easiest ways to get involved in online sales globally [5]. There are various factors influencing the readiness of SMEs to implement information technology, including the optimism, knowledge, and skills of SME actors, while the inhibiting factors include perceptions of discomfort and reluctance to change. Research conducted by Firmansyah et al. [6] found that the perception of the use of information technology (IT) depends on the factors of optimism and innovation. They also concluded that (1) insecurity is a limiting factor, (2) the perception of the use of IT is a factor determining the intention to use IT, and (3) SME actors need to maintain a positive attitude in understanding IT.

Research conducted by Kurniati and Prajanti [7] states that the role of batik entrepreneur owners is very large in innovation, which includes product innovation, marketing innovation, and business alliances. The role of entrepreneurship in marketing innovation has the highest elasticity concerning the production and sales of batik, followed by the roles of product innovation and business alliances. In a competitive industrial environment, batik industry entrepreneurs have an important role in improving the economy of the company and its industry. Innovation is not limited to large-scale enterprises, which generally have a research and development (R & D) division, but also includes small businesses such as batik SMEs, which also require innovation activities [4,6–10]. SMEs benefit from organizational flexibility in responding to environmental changes, but most SMEs have drawbacks in access and innovation capacity due to limited resources and economies of use [11].

Rosenberg [1] stated that over the past thirty years, virtual reality (VR) and augmented reality (AR), as immersive technologies, have continued to evolve as sophisticated forms of technology, enabling users to develop various business processes using VR and AR. During the same period, the demand for network speed has continued to increase progressively. These advances increase the prospects for the worldwide adoption of VR and AR. The metaverse, as a platform provider that can combine VR and AR, has begun to be widely used in the business world and the world of education to develop networks and promote their products. The increase in the use of the metaverse has had a huge impact on society. It is very important to consider the risks and prepare appropriate regulations so that the negative aspects of using information technology can be minimized.

This study was designed based on the practice gap between small and medium enterprises (SMEs) facing problems related to data volumes that are too large, a lack of information, and a lack of knowledge. In fact, to be able to develop better, SMEs must be able to make timely decisions. On the other hand, the metaverse is associated with high-level technologies that take advantage of VR and AR together, which brings many advantages and benefits to businesses. The technology gap is very clear in this regard, and the capabilities of SMEs must be developed in such a way as to allow a technological pursuit of the metaverse [12].

Lasem is a small area in the form of a sub-district in Rembang Regency, Central Java, which is associated with the classical style of coastal batik craft. Lasem Batik is rich in motifs and colors due to the accumulation of local culture (Javanese) influenced by immigrant culture, mainly derived from Champa (Vietnam), India, China, and the Netherlands. The main characteristics of Lasem's hand-drawn batik lie in the color display in the form of a combination of bright colors, such as red (bang-bangan), blue, yellow, and green, which is different from other forms of coastal batik [13]. Lasem Batik will continue to prosper, in addition to marketing and tourism development. Lasem, known as Little China, also has the potential to develop its tourism sector.

Lasem Batik's marketing has experienced a great deal of turmoil and is greatly affected by the economic conditions. Efforts to conduct marketing through offline and online channels have been made. Marketing with an online model is achieved, among other things, by performing live streaming through Instagram. The pandemic conditions and changes in transportation routes between cities and between provinces, which were moved to toll roads, no longer passing through Lasem, greatly affected the marketing of Lasem Batik products. Efforts made by Lasem Batik include more aggressive marketing through Facebook, WhatsApp, and Instagram. The originality of this research is the investigation of the possibility of Lasem Batik SMEs conducting marketing through the metaverse.

## 2. Literature Review

### 2.1. Metaverse

"Metaverse" has different definitions depending on the context in which it is described. Generally, the metaverse is related to virtual reality (VR) and augmented reality (AR). The metaverse is a persistent and immersive world of simulation experienced in the first person, as well as in a large group of users simultaneously present in a large space that becomes a completely virtual environment (virtual world), or it can exist as layers of virtual content overlaid in the real world with convincing registration [1]. Virtual reality (VR) is an immersive and interactive simulation environment experienced in the first person and offers the user a strong sense of presence. Mystakidis [14] defined the metaverse as a post-reality universe, a perpetual and persistent multiuser environment merging physical reality with digital virtuality. It is based on the convergence of technologies that enable multisensory interactions with virtual environments, digital objects, and people, such as virtual reality (VR) and augmented reality (AR). Laeeq [3] described the concept of a fully immersive virtual world where people gather to socialize, play, and work. It is a simulated digital environment that combines augmented reality (AR), virtual reality (VR), blockchain, and social media principles to create areas for rich user interaction that imitate the real world. Based on the above definitions, it can be concluded that the metaverse is a persistent simulated world that combines augmented reality (AR), virtual reality (VR), blockchain, and social media principles to create an area for interaction between platform users and allows users to simulate the real world.

### 2.2. Trade in the Metaverse

Trading in the metaverse is relatively new and yields quite promising results when businesses make use of its advantages and obtain good outcomes. For example, virtual real estate company Gucci, a fashion brand, is collaborating with game developer Roblox to sell products in the metaverse. Balenciaga partnered with Epic Games, the creators of Fortnite, to provide a virtual boutique. RTFKT, a well-known metaverse brand with a collection of shoes, was purchased by Nike to expand its sales. An 18-year-old designer sold more than USD 3 million in virtual shoes in less than seven minutes. Nike is looking to hire virtual apparel designers and trademark apps [3].

The whole world is experiencing a significant shift from the actual economy to the digital economy. Work and life are increasingly dependent on the internet because various activities depend on information obtained from online systems. The internet is often the main gateway for millions of people to interact and socialize with each other, sell products,

and transact through a virtual environment; long distances are not an obstacle. Technology is key to the maintenance of many jobs. The demand for virtual reality is growing along with the industry that utilizes the metaverse [15]. In the metaverse, a virtual world that transcends reality, artificial intelligence, and blockchain technology are combined. With new technologies related to the development of computers, graphics, and hardware, cyberspace has become a reality. More and more digital asset transfers will occur on the blockchain via avatars. It is expected that the digital value paradigm will form a new economic model.

Soepeno [16] states that the metaverse is a 3D virtual augmented reality where anyone can experience anything in a virtual environment and connect. The use of the application in an extended and tactical virtual environment, meeting people in virtual reality, engaging in physical activities in a virtual holographic metaverse, and many more opportunities and activity options are offered.

### 2.3. Technology Acceptance Model

The Technology Acceptance Model (TAM) was developed by Davis [17]. It explains how the acceptance of technology by users will affect the use of the technology itself. This theory was adapted from several models built to analyze and understand the factors that influence the acceptance of using new technologies. The Technology Acceptance Model is part of the Theory of Reasoned Action, which is most widely used by researchers to explain the reasons for using information technology [15,16,18]. TAM is a theory designed to explain how users apply and understand information technology. TAM aims to explain and predict the acceptance of information relations with technology-based users. In addition, TAM is also used to explain the behavior of end users with variations, and the number of user populations is increasing. TAM has three key variables: perceived usefulness (PU), perceived ease of use (PEU), and behavioral intention to use (BIU). Perceived usefulness and ease of use of IT are the most significant factors influencing the desire to adopt information technology.

Since its introduction by Davis [17], TAM has been widely used by researchers to explain the user acceptance of technology. The model explains that if a technology is deemed useful, the technology will be adopted, which seems to be supported by the PEOU. There is a positive and strong correlation between the acceptance and use of technology variables and user satisfaction. The results of this study are useful not only for managers but also for manufacturers, technical support, online support, and after-sales services because they are advised to develop strategies for user satisfaction [19].

Davis [17] developed and validated a new scale for two specific variables, perceived usefulness and perceived ease of use (see Figure 1), which were hypothesized to be fundamental determinants of user acceptance. Davis' main goals were as follows: first, the theory presented should be able to enhance our understanding of the user acceptance process, providing new theoretical insights into the design and implementation of successful information systems; second, TAM must provide a theoretical basis for the testing of acceptance among users of the system.

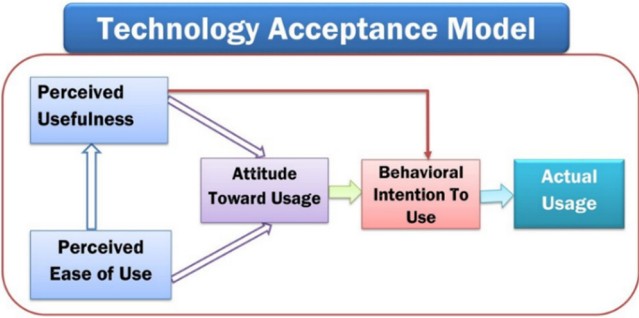

**Figure 1.** TAM. Source: Davis (1989).

Davis [17] found that usability has a much greater correlation with user behavior than ease of use. Perceived ease of use may, in fact, be a causal antecedent for perceived usefulness; that is, PEOU affects technology acceptance (TA) indirectly through PU. In the last decade, TAM has received much attention and empirical support (see Gefen and Straub [20]; Venkatesh [21]). It is estimated that around 100 studies were published in journals, proceedings, or technical reports related to TAM between 1989 and 2001.

*2.4. Theory of Planned Behavior*

The Theory of Planned Behavior was developed by [22] and is often used to explain individual behavior associated with the use of information technology. Many studies have proven this and provide empirical evidence for researchers. Intention to behave with different types of technology can be predicted with high accuracy from attitudes towards behavior, subjective norms, and perceived behavioral control. Nita considers this in conjunction with perceived behavioral control, explaining the considerable variation in actual behavior. Huang and Chen [23] stated that the Theory of Planned Behavior developed by [22] has become the main theory used to explain actual intentions and behavior in various fields. The Theory of Planned Behavior considers attitudes, subjective norms, and perceived behavioral control (PBC) as three useful factors in predicting actual intentions and behavior.

**3. Research Methods**

The sample of this research consists of the owners of Lasem Batik SMEs in the middle-to-upper business category whose sales have reached the provincial, national, and international levels. The criteria for selecting this sample referred to the Indonesian Government's Law No. 20 of 2018 [24] regarding the following criteria for SMEs: (1) a minimum net worth of IDR 500 million, and (2) a sales turnover in one year of at least IDR 2.5 billion. Based on these criteria, the selection of SMEs was conducted through the Lasem Batik entrepreneur cooperative. There are 79 batik entrepreneurs who are members of the batik entrepreneur cooperative. The selection was carried out on 79 batik entrepreneurs, and 40 batik business owners who met these criteria were obtained.

This study uses an interpretive phenomenological analysis (IPA) approach, which provides the researcher with the best opportunity to understand in depth the research participants' experiences of the metaverse. As a 'participant-oriented' approach, the interpretive phenomenological analysis approach allows the research participants to express themselves and provide their recollections of their experiences as they see fit, without distortion and/or prosecution. The primary aim of the IPA approach is to explore the 'hands-on experience' of the research participants and to enable them to generate research findings through their experiences [25]. Qualitative research methods impart an added advantage to exploratory abilities. Qualitative methodologies enable researchers to advance and apply their interpersonal skills and subjectivity to their exploratory research process. In a study with an interpretive phenomenological analysis (IPA) approach, IPA provides the best opportunity for researchers to understand the details of the 'life experiences' of research participants in depth.

The research process was conducted as a focus group discussion, which started with an explanation of the metaverse and the various functions of the metaverse. The process of applying the metaverse for marketing in Lasem Batik was also conducted, with the assistance of various workshops on materials related to creating a metaverse space, photography, branding, and promotion through a metaverse. The focus group discussion focused on understanding the metaverse, the readiness of the owners of Lasem Batik to consider the metaverse, the obstacles and challenges that will occur, and the prospects for marketing through the metaverse.

## 4. Results and Discussion

### 4.1. The Understanding of the Metaverse

A focus group discussion was conducted to discuss the possibility of using the metaverse to increase the marketing of Lasem Batik. The initial question in this research was regarding the participants' understanding of the metaverse. In the focus group discussion, the researcher explained in detail what the metaverse is, with the various functions and uses of the metaverse. Participants were asked to respond to questions about their understanding of the metaverse and the likelihood that they would use the metaverse to expand their marketing channels. The results of the focus group discussion are depicted in Figure 2.

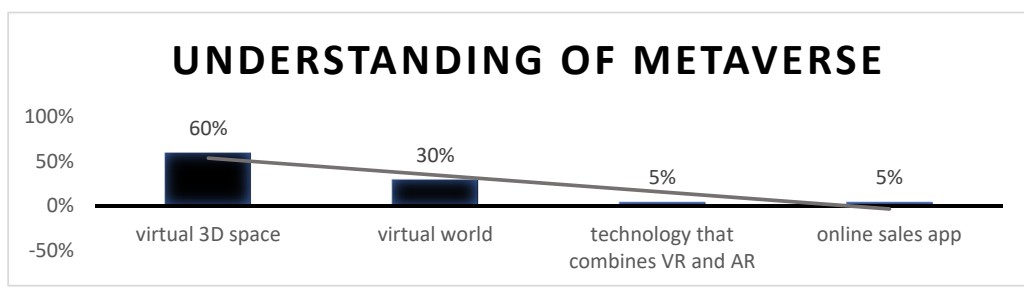

**Figure 2.** The understanding of the metaverse. Source: processed primary data, 2022.

Based on Figure 2, the majority of participants stated that the metaverse is a virtual 3D space. This understanding was deepened by discussions about virtual 3D spaces that combine VR and AR, which today is often referred to as the metaverse. The metaverse can be used by various institutions and associations to conduct online meetings and can even be used to market products, such as through model exhibitions. In the acceptance model theory, this stage involves explaining how technology is accepted by users. The acceptance of technology begins with an understanding of the technology itself [3,22,23].

The introduction of the metaverse technology was conducted by providing examples of various batik exhibitions conducted with AR and VR technologies. The control of metaverse technology towards Lasem Batik entrepreneurs can be seen in Figure 3. This exhibition was designed to include various activities that were photographed and videotaped in the metaverse space. Exhibitors could observe various models of batik, and, if interested, they could perform transactions with batik owners.

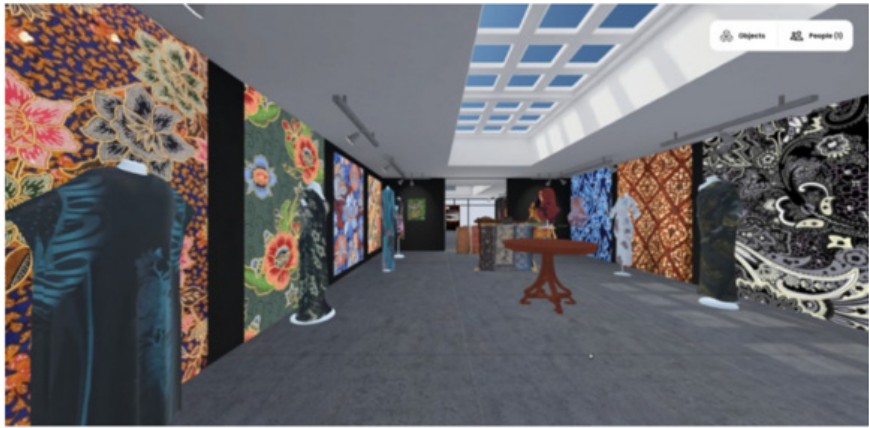

**Figure 3.** Batik exhibition in the metaverse. Source: workshop result, 2022.

### 4.2. Intended Use of the Metaverse

The participants were asked what they expected to achieve by using the metaverse, and 43% stated that by using the metaverse they would obtain access to a virtual community (see Figure 4), which could provide various benefits. In addition, 33% of participants stated

that using the metaverse would create a growing trade, and 25% of participants stated that they would obtain familiarity with a different world. We can conclude that batik business owners hope that by using the metaverse, they will be able to achieve many benefits, especially related to market expansion, which is quite different from the brand's past experiences.

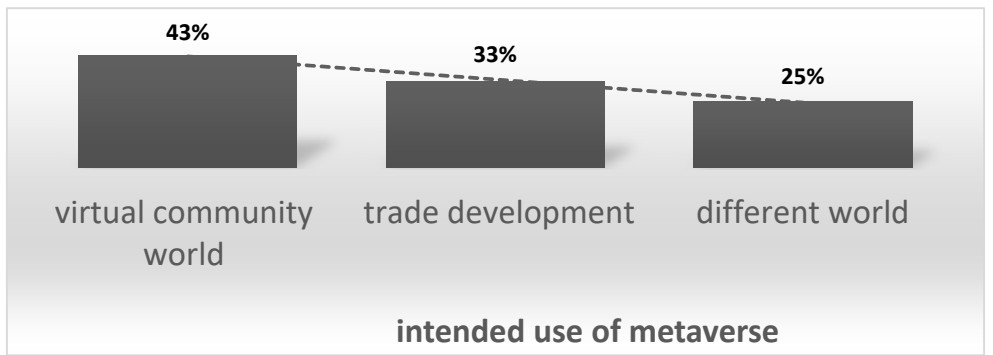

**Figure 4.** Intended use of the metaverse. Source: processed primary data, 2022.

Based on the responses of the batik entrepreneurs who were invited to discuss their attitudes towards using the metaverse, they showed great interest in expanding their marketing channels to a wider area. Lasem Batik entrepreneurs also accept the challenge of studying the metaverse with new knowledge and techniques never considered before. Great interest in using the metaverse is one of the keys to successfully using this type of information technology. It has been explained that TAM has three main variables: perceived usefulness (PU), perceived ease of use (PEU), and behavioral intention to use (BIU) [15–18]. Interest in using the metaverse is a determining factor in whether or not this information technology platform is used. This is also supported by the behavioral control conducted by the researchers by providing examples of the use of the metaverse and descriptions of development opportunities that can be explored. Such conditions align with the conditions described in the Theory of Planned Behavior [15–18].

In its development, the process of implementing the metaverse to expand the marketing channels for Lasem Batik was well-received by the local community. This is shown by the large number of mass media interested in becoming involved, particularly by publicizing training activities and workshops related to the metaverse conducted by researchers at Lasem [26–29]. Marketing activities constitute an interaction between the environment and the companies, which are self-centered during their operations. Such activities have included functional, transactional, competitive, mixed, integral, and relational aspects, among others [30].

### 4.3. Constraints of Using the Metaverse

Aware of the current condition, whereby batik business owners are not fully familiar with the metaverse or the various tools needed, we asked whether any obstacles were expected to greatly affect the use of the metaverse platform as a marketing tool for Lasem Batik. Participants' responses were as follows: 48% of Lasem Batik owners stated that they lacked knowledge about the metaverse (see Figure 5), and 8% of participants stated that they would refuse to use the metaverse because their industry adheres to traditional practices. Meanwhile, 10% of participants stated that the main obstacle faced was that the infrastructure required to use the metaverse did not yet exist. In addition, 15% of participants stated that they had to improve their human resources and were willing to perform marketing through the metaverse. Finally, 20% of participants stated that they required high creativity in marketing their products through the metaverse platform, while their current business processes were focused on the patterns and development of batik motifs.

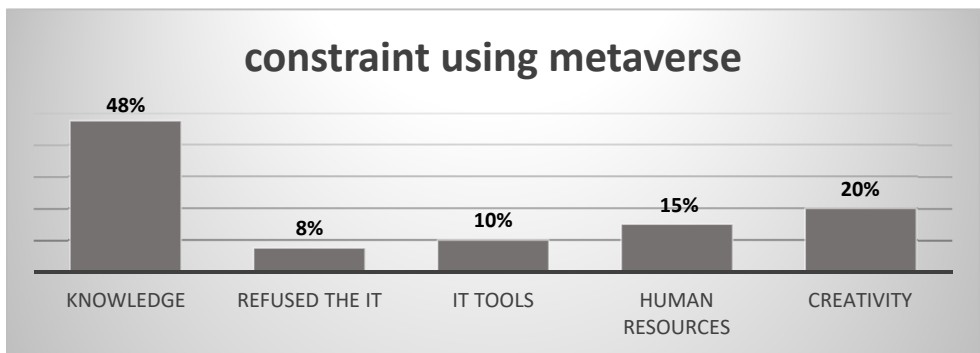

**Figure 5.** Constraints of using the metaverse. Source: processed primary data, 2022.

The obstacles faced by Lasem Batik entrepreneurs in using the metaverse are not fully considered to be inhibiting factors in developing their knowledge and marketing channels. Although they are still unfamiliar with the metaverse and the tools used to access the metaverse, the high willingness to learn among these batik entrepreneurs could help them to overcome the obstacles that they face. As revealed by [29], Lasem Batik entrepreneurs have established a cooperative formed to develop their businesses with various activities, including savings and loans, workshops, and discussions on batik development. In this cooperative, many problems have been resolved, including the problem of expanding the batik marketing channel.

*4.4. The Possibility of Using the Metaverse Platform as a Marketing Tool*

In general, the enthusiasm to participate in various metaverse-use preparation programs is extremely high. Based on the analysis of the responses of batik entrepreneurs when asked about the possibility of using the metaverse to market their batik, the answers were as follows: 13% of participants said they would not use the metaverse platform, their main reason being there is no evidence that the metaverse can increase sales; 48% of participants stated that they would use the metaverse platform; another 40% were still hesitant to use the metaverse platform for marketing their products.

The considerations were the same as those of the participants who answered "no"—namely, there is no convincing evidence that marketing with the metaverse can be achieved and can increase sales. The condition of batik entrepreneurs who are still unsure is in accordance with the research results of Hastuti et al. [31], which states that Lasem Batik entrepreneurs have not used information technology as a tool in managing the business. This is illustrated in Figure 6. These results support what is stated in the Technology Acceptance Model. There is a positive and strong correlation between the acceptance and use of technology and user satisfaction. The results of this study are useful not only for managers but also for manufacturers, technical support, online support, and after-sales services, as they are important in the development of user satisfaction strategies [23].

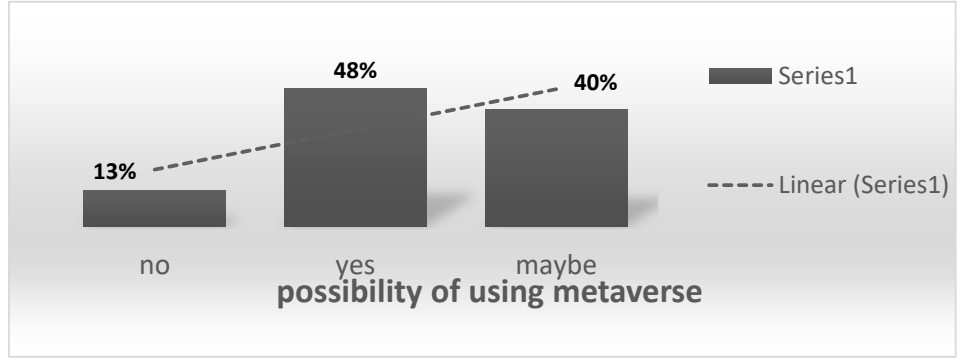

**Figure 6.** The possibility of using the metaverse platform. Source: processed primary data, 2022.

The involvement of Lasem Batik entrepreneurs in activities designed to prepare them for using the metaverse platform is quite high. Preparations include creating a metaverse platform, preparing a space for exhibitions, and creating metaverse assets. To expedite the process of placing batik assets in the metaverse, a photography workshop was held. This workshop was intended to document Lasem Batik's work in the metaverse. In addition, a workshop was also held on exploring batik branding and promotion on the metaverse platform. With these workshops, it was hoped that the Lasem Batik entrepreneur could increase his understanding of the metaverse to overcome his concerns about using it. In addition, a metaverse platform prototype was also developed for the Lasem Batik exhibition. Several ready-made metaverse room templates were provided for Lasem Batik SMEs to make it easier for them to utilize the technology for their business. SMEs can use the room templates provided by the platform and collaborate with other SMEs to hold metaverse exhibitions to attract global customers. This is consistent with what is stated in the Theory of Planned Behavior. The Theory of Planned Behavior was developed by [17] and is often used to explain individual behaviors related to the use of information technology.

*4.5. The Hope after Knowing the Metaverse*

It was hoped that the workshop provided as part of the preparatory process for using this platform could provide deeper and more detailed insights for Lasem Batik business owners. During the workshop process, in addition to providing knowledge and skills enrichment materials about the metaverse, an activity evaluation and analysis of increasing interest was also conducted using the metaverse platform. Based on the evaluation results, it was found that 75% of participants would continue to use the metaverse platform as an alternative to the Lasem Batik marketing platform and 25% of participants stated that their use of the metaverse platform required assistance in its operation (see Figure 7). Based on this evaluation, it can be concluded that 100% of participants would use the metaverse platform as an alternative method of marketing batik products. Based on the results, it can also be concluded that the metaverse workshop and its supporting tools, provided by the researchers, encourage batik entrepreneurs to use the metaverse platform. Participants who are hesitant at first are encouraged to become involved in using the metaverse, even if they require assistance initially.

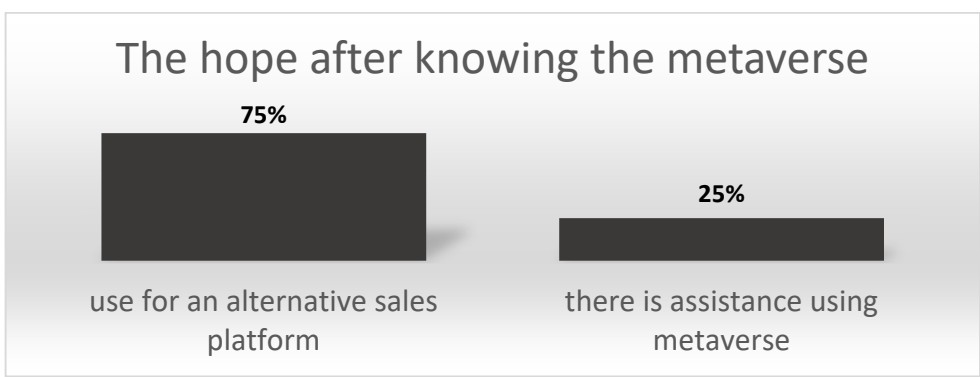

**Figure 7.** The hope of using the metaverse. Source: processed primary data, 2022.

*4.6. Environmental Support*

The batik business can develop well if there is support from various parties, such as family members, the surrounding community, and the government. Government support in the form of exhibitions and events will increase product marketing and encourage the emergence of batik product innovation. This is possible because the exhibitors could visit each other and discuss the uniqueness of each of their products. Government support can also be provided in the form of skills development and training for batik craftsmen [32]. Batik has been recognized as one of the products developing rapidly in the creative economy

in Indonesia. The batik trade has grown and developed in Indonesia as a form of preserving regional cultural wealth, which has been developed as a form of local wisdom [33].

Recently, the central government, through the village development program, has formed a tourist village in an effort to develop regional potential and provide a place to foster community creativity. Tourism villages can also increase the incomes of local entrepreneurs. The community's strong support for the creativity of its citizens—for example, the creativity in producing batik and other food businesses—is achieved by purchasing these products. This will increase the turnover of the village economy. In addition, batik entrepreneurs can also encourage their employees to take part in the training provided so that they can jointly develop their own abilities and support the batik business in which they work. Small and medium enterprises (SMEs) face unique challenges in the business environment. SMEs must be able to keep up with change if they are to survive and grow and meet the expectations of creating investment and job opportunities. SMEs are expected to successfully adapt to technological changes and advances, customer expectations, supplier requirements, the regulatory environment, and increasing competition [34–36].

The government must ensure stable macroeconomic conditions, especially regarding exchange rates, interest rates, and inflation. Policies for SMEs must be able to protect and support their businesses in times of low economic growth. Governments can increase the promotional channels for new business development through intensive mentoring and removing known barriers. SMEs are introduced to an integrated multi-sector SME strategy. They are provided with partnership facilitation, entrepreneurship promotion and support, ease of access to financing, a focus on export promotion, and competitiveness development [22,23].

In this study, the community's support for batik businesses' efforts to expand their marketing through the metaverse platform was perceived by batik business owners. This support is depicted in Figure 8 as follows: 65% of participants stated that they received full support from the community, 13% of participants stated that they encountered obstacles, and 23% of participants wished to learn more about the metaverse. This indicates a society open to new developments that will advance the region.

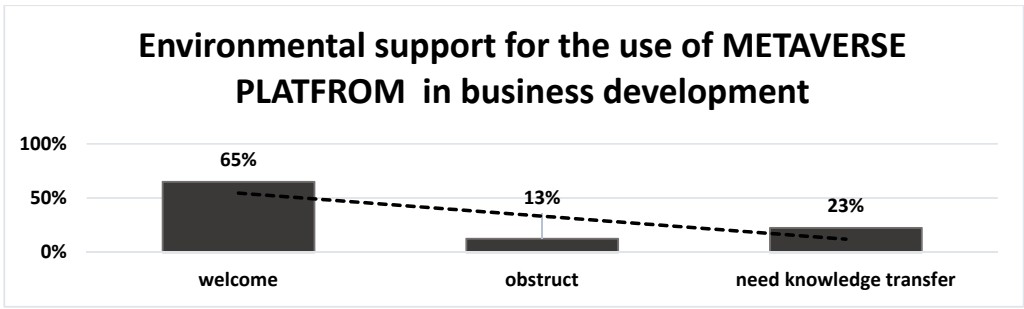

**Figure 8.** Environmental support for the use of the metaverse platform in business development.

*4.7. Continuity of Business Planning*

Continuous business plans and strategies provide effective solutions for multi-cloud and microservice approaches. A business continuity plan can help to provide a backup and measures for disaster recovery. In the business plan, actions have been raised to ensure sustainable business processes during disasters and emergencies. Business continuity planning methods include risk assessment, impact analysis, and a full business continuity strategy [37].

The talent for making Batik that has been owned by the people of Lasem and combined with the cultural heritage program by the government has made Lasem Batik continues to grow to this day. Cultural and Batik observers from universities and non-governmental organizations also coloured the development of Lasem Batik [38].

The ups and downs of Lasem Batik occur because the main thing is the willingness of the Lasem people themselves to keep the cultural heritage growing in Lasem or not.

Many Lasem Batik businesses have gone out of business because there is no successor to their business. The heirs are more likely to work in other sectors and outside Lasem. This growing awareness of the importance of the Lasem Batik business being maintained and developed is a crucial factor that the Batik community and the Rembang district government as well as cultural observers realize so that until now Lasem Batik is getting better and better [39].

Based on the focus group discussions with batik business owners (results can be seen in Figure 9), we obtained the factors that influence the growth of Lasem Batik. Focus group discussion participants stated that the factors that determined the sustainability of Lasem Batik were as follows: 28% of participants stated that they maintained the quality of their products, 23% of participants stated that they conducted continuous promotions both online and offline, and 18% of participants stated that the continuity of the batik business was due to the creativity and innovation of batik owners, who continued to grow. Moreover, 3% of participants stated that to develop their business they had to add employees, while 9% of participants stated that they had to use the opportunity to develop as much as possible. Meanwhile, 3% of participants stated that they had to maintain the traditional nature of Lasem Batik, and 15% of participants stated that they had to always maintain good relations with customers and expand their relationships.

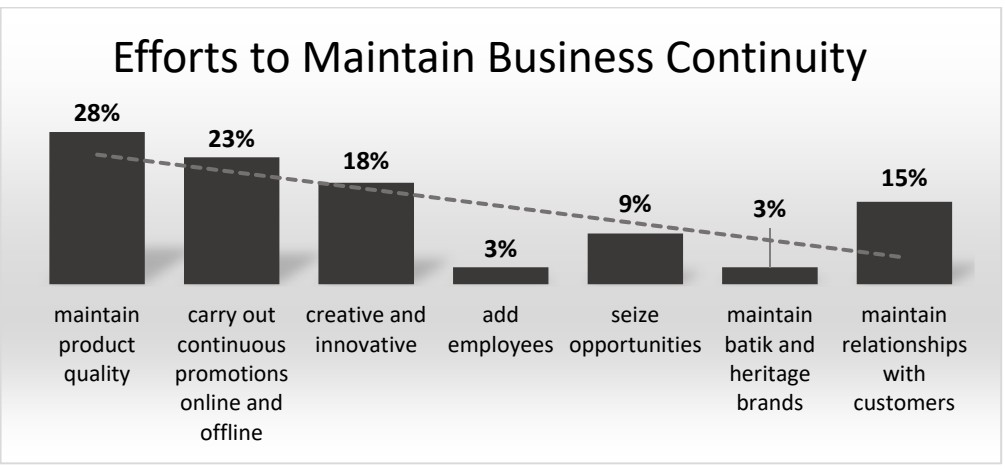

**Figure 9.** Efforts to maintain business continuity.

Lasem Batik has developed over time and has experienced some turmoil. The cultural value in Lasem continues to be developed by the local community and government [40]. The Lasem community's talent for batik, combined with the government's cultural heritage program, has allowed Lasem Batik to continue to grow to this day. Observers of culture and batik from universities and non-governmental organizations have also promoted the development of Lasem Batik [41,42].

The turmoil and success experienced byLasem Batik has occurred because of the willingness of the Lasem people themselves to maintain the cultural heritage that has developed in Lasem. This is the main requirement. Many Lasem Batik businesses have become bankrupt because there are no successors. Successors are more likely to work in other sectors and move outside Lasem. The growing awareness of the importance of the Lasem Batik business to be maintained and developed is a crucial factor that the batik-making community should be aware of. This can be strengthened by the support of the Rembang District government, as well as culturalists around Lembang, or even by national culturalists paying more attention to the development of Lasem Batik so that Lasem Batik can continue to grow and expand.

## 5. Conclusions

Studies on the readiness of SMEs in Lasem Batik for using the metaverse as a marketing channel have not been conducted widely. The originality of this research is its focus on

metaverse applications in SMEs. Thus far, the metaverse has been used in large institutions, both in the world of business and the world of education. This study describes the readiness of Lasem Batik SMEs to utilize the metaverse by adopting an interpretive phenomenological analysis (IPA) approach. The phenomenological analysis provides researchers with an opportunity to understand, in-depth, the research participants' experiences of the metaverse. The study results show that the readiness of batik entrepreneurs to use the metaverse is highly dependent on the support of various parties. A strong desire to progress and develop one's business is the main factor determining one's intention to use the metaverse. Support from other parties, especially those who provide assistance regarding new technologies that businesses have never used before, is urgently needed by batik SMEs. The results of this study prove that information technology assistance can eliminate doubts about using the metaverse and promote strong beliefs to encourage involvement and the exploitation of opportunities to develop the business. The involvement of the mass media and non-governmental organizations is also a factor that can provide support for the intention of SMEs to use the metaverse. With mass media publications, it will be possible to increase the news about batik itself; this will raise the prestige of batik in the eyes of buyers and prospective buyers. Lasem Batik will also be increasingly recognized by many people. From mass media publications, we may also attract the government's attention to assist in the development of batik SMEs.

Implementing the metaverse for batik marketing includes creating a metaverse platform, preparing a space for exhibitions, and creating metaverse assets. To expedite the process of placing batik assets in the metaverse, a photography workshop was held. A workshop was also held exploring batik branding and promotion on the metaverse platform. With these workshops, it was hoped that the Lasem Batik entrepreneur would increase their understanding of the metaverse so that they could overcome their concerns about using the metaverse.

The main obstacle faced by Lasem Batik entrepreneurs in using the metaverse is their fear of not being able to use the tools to access the metaverse. In addition, the sustainability of using the metaverse depends on the ability of batik entrepreneurs to obtain the optimal benefits from the metaverse. Another challenge facing batik entrepreneurs is that SMEs must be able to keep up with change if they are to survive and grow and meet expectations by creating investment and job opportunities. SMEs are expected to successfully adapt to technological changes and advances, customer expectations, supplier requirements, the regulatory environment, and increasing competition.

Based on the results of this study, researchers can implement the use of the metaverse among Lasem Batik entrepreneurs and cooperate with the local government based on the Lasem Batik SMEs' characteristics. As a result of the research, a prototype of a metaverse platform for a Lasem Batik exhibition has been developed. Several ready-to-use metaverse room templates have been provided for the Lasem Batik SMEs to ease the process of utilizing the technology for their business. The SMEs can use the room template provided by the platform and join other SMEs to hold a metaverse exhibition to attract global customers. These rooms will be connected to create a metaverse exhibition to attract global customers. Although the platform can not only be viewed using a metaverse headset but also can be accessed by laptop users or smartphone users, the most immersive experience can be achieved by the use of a metaverse headset. Batik cloth and batik outfits that are displayed in 3 dimensions in the immersive room and can be seen in detail will create a new experience for the customers who see those clothes. The new experiences that customers feel can increase better interest in batik cloth and batik outfits. The development of exhibition spaces within the metaverse can be an alternative to displaying Lasem Batik products virtually and regularly in various countries that the SMEs have never visited before. From the results presented, future research could analyze the impact of the metaverse on the marketing of Lasem Batik by measuring the increase in sales, both in terms of the quantity and quality of its customers.

## 6. Recommendations

Recommendations are provided for Lasem Batik entrepreneurs. They can use the results of this study, which show that strong intentions accompanied by support and assistance will help to strengthen the use of information technology as one of the channels that will expand the range of sales of Lasem Batik products. Lasem Batik entrepreneurs can use this research as a reference to ensure that their knowledge is up to date, especially so that the batik that they produce can reach a global level. The batik business can thus grow rapidly because it can keep up with the latest marketing developments but can also anticipate the risks that will occur. Innovation and creativity continue to be developed as one of the keys to business success. The main recommendation for the government is that they must continue to support Lasem Batik SMEs so that the cultural heritage can be well maintained and develop into a national asset that is known internationally. Recommendations for future research are that research can be conducted using quantitative methods that examine the impact of using the metaverse to improve financial performance, increase Lasem Batik branding, and expand its marketing.

**Author Contributions:** Conceptualization, T.D.H. and R.S.; methodology, T.D.H.; validation, R.S.; formal analysis, T.D.H.; investigation, T.D.H.; resources, T.D.H.; data curation, F.K.; writing—original draft preparation, T.D.H.; writing—review and editing, R.S.; visualization, F.K.; supervision, R.S.; project administration, T.D.H. and F.K.; funding acquisition, T.D.H., R.S. and F.K. All authors have read and agreed to the published version of the manuscript.

**Funding:** This research and APC was funded by the Indonesian Ministry of Education, Culture, Research, and Technology.

**Data Availability Statement:** All data has been present in main text.

**Acknowledgments:** This article was a part of a research project supported by the Indonesian Ministry of Education, Culture, Research, and Technology under the scheme of the Matching Fund Kedaireka in 2022, titled Metaverse-Based Batik Event Organizing Platform for Increasing and Expanding Lasem Batik Sales Channels.

**Conflicts of Interest:** The authors declare no conflict of interest.

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
