# Peer review of "The Readiness of Lasem Batik Small and Medium Enterprises to Join the Metaverse"

_computers, doi:10.3390/computers12010005_

Round 1

Reviewer 1 Report

Dear authors,

Your paper is very interesting and I do hope that my comments will make it even better.

- Pls check the journal rules on how to reference in the paper -  check the journal template

-in the research background, it would be nice to have introduction to the paper

-for some terms in the literature review part references are missing when they are defined

- also for the examples in chapter 2.2. references are missing

-on page 3 you are stating source Davis (1989a) and on page 4 below Figure 1 is source Davis (1989)- is this the same or different?

-line 149 - you state many studies but no references to the studies

-in the research methods chapter - pls elaborate how did you select the sample

-you need to upgrade your discussion since it uses only elementary statistics and it is first shown on the Figures and the elaborated in a text (repeated)

-in conclusion you are again repeating research results

- Reccomendation part is hard to understand

- what to do next regarding the research? This is also missing

Author Response

Thank you for the review given, here are the revisions we have made. Please see the attachment

Reviewer 2 Report

1) In my opinion, the research structure and the adopted methodology are not appropriate for drawing conclusions in a scientific article. In my opinion, the article in its current form requires too many corrections and does not meet the standards of scientific articles. In my opinion, the results presented here can be presented to the company's management, they can be a white paper for the internal needs of the organization. I do not see a relationship between TAM and the results of the research, the discussion of the results and the conclusion.

2) The authors did not show a research gap. The research gap should be supported by literature. What is the purpose of the work? What are the regional or international recommendations? In my opinion, as it stands, the article is only local in nature and resembles an "internal research" report.

3) Figure 1. TAM. Source: Davis (1989) – In my opinion, the description of this figure is too short. Moreover, the quality of this figure is inadequate. The graphic quality of this figure should be improved - that is my recommendation.

4) Section "5. Conclusion" contains neither conclusion nor summary. Currently, this section contains the test results, listed point by point. In my opinion, this section needs thorough improvement.

5) The "6th Recommendation" section only contains a few general conclusions and general wording. This section does not contain any specific recommendations for public administration units or marketing managers. This section requires a thorough improvement. Recommendations should result directly from the research results.

Author Response

(The authors gave the same response as above.)

Round 2

Reviewer 1 Report

Dear authors,

I can see a significant improvement of your paper but I still see a room for improvement

1. check English since on several place there are sentences which are hard to understand

2. regarding methodology - how many entrepreneurs met the criteria and how did you choose these 40? Intentionally or randomly?

3. Check the theory names and how you write them

4. Again in the conclusion you are stating what you have done - workshops, --- but what will you do next as a researcher is still missing

Reviewer 2 Report

In my opinion, the abstract should be improved. In its current form, the abstract does not meet the standards of a scientific article. According to the guidelines of the Computers journal (ISSN 2073-431X): The abstract should be a single paragraph and should follow the style of structured abstracts, but without headings: 1) Background: Place the question addressed in a broad context and highlight the purpose of the study; 2) Methods: Describe briefly the main methods or treatments applied. Include any relevant preregistration numbers, and species and strains of any animals used. 3) Results: Summarize the article's main findings; and 4) Conclusion: Indicate the main conclusions or interpretations. The abstract should be an objective representation of the article: it must not contain results which are not presented and substantiated in the main text and should not exaggerate the main conclusions.

The authors of the article did not dispel my doubts regarding the research methodology. In my opinion, the article presents the results of an internal investigation/discussion and is popular in nature. The research results have a local reference. In my opinion, the authors of the observations/ interviews did not make a new contribution that would allow us to better know and understand the Metaverse. In addition, I have doubts whether the opinion-forming articles (opinion type paper) are appropriate for a computer science journal, especially the thematic section, which aims to present the latest research in the field of augmented and virtual reality.

Round 3

Reviewer 2 Report

I appreciate the work the authors put into improving the article. The substantive value of the article has been corrected. The article requires some corrections, but they are mostly of an editorial nature and concern the figures: 1) I still argue that figure 1 does not meet the quality standards (too low resolution in px; wrong font, etc.),

2) Figure 4 some values in the graph are not visible (they are cut off), at least in the file to which I have access as a reviewer;

3) Figure 6 is signed twice the figure has a signature above and below the figure. The signature is the same it is duplicated. Two of the same signatures do not bring anything new. Is the figure 6 even needed?
